# A mechanistic model for long-term immunological outcomes in South African HIV-infected children and adults receiving ART

Eva Liliane Ujeneza[1,2]*, Wilfred Ndifon[2], Shobna Sawry[3], Geoffrey Fatti[4,5], Julien Riou[6], Mary-Ann Davies[7], Martin Nieuwoudt[1,8], IeDEA-Southern Africa collaboration

[1]Department of Science and Technology and National Research Foundation, South African Centre for Epidemiological Modelling and Analysis (SACEMA), Stellenbosch University, Stellenbosch, South Africa; [2]African Institute for Mathematical Sciences (AIMS), Next Einstein Initiative, Kigali, Rwanda; [3]Harriet Shezi Children's Clinic, Wits Reproductive Health and HIV Institute, Faculty of Health Sciences, University of the Witwatersrand, Johannesburg, South Africa; [4]Kheth'Impilo AIDS Free Living, Cape Town, South Africa; [5]Division of Epidemiology and Biostatistics, Department of Global Health, Faculty of Medicine and Health Sciences, Stellenbosch University, Cape Town, South Africa; [6]Institute of Social and Preventive Medicine, University of Bern, Bern, Switzerland; [7]Centre for Infectious Disease Epidemiology and Research, School of Public Health and Family Medicine, University of Cape Town, Cape Town, South Africa; [8]Institute for Biomedical Engineering (IBE), Stellenbosch University, Stellenbosch, South Africa

**Abstract** Long-term effects of the growing population of HIV-treated people in Southern Africa on individuals and the public health sector at large are not yet understood. This study proposes a novel 'ratio' model that relates CD4+ T-cell counts of HIV-infected individuals to the CD4+ count reference values from healthy populations. We use mixed-effects regression to fit the model to data from 1616 children (median age 4.3 years at ART initiation) and 14,542 adults (median age 36 years at ART initiation). We found that the scaled carrying capacity, maximum CD4+ count relative to an HIV-negative individual of similar age, and baseline scaled CD4+ counts were closer to healthy values in children than in adults. Post-ART initiation, CD4+ growth rate was inversely correlated with baseline CD4+ T-cell counts, and consequently higher in adults than children. Our results highlight the impacts of age on dynamics of the immune system of healthy and HIV-infected individuals.

*For correspondence:
ujeneva@gmail.com

Competing interests: The authors declare that no competing interests exist.

## Introduction

The efficacy with which antiretroviral therapy (ART) suppresses HIV viral load and restores lymphocyte responses to pathogen-derived antigens, normally lost due to viral replication, is well established (*Cooney, 2002*; *Autran et al., 1997*). Following ART initiation, for the majority of HIV-infected individuals, the CD4+ T-cell counts rapidly increase for approximately 4–6 months (*Lawn et al., 2006*), followed by a slower increase in the next 2–4 years, after which cell numbers plateau (*Pinzone et al., 2012*). Despite the successes of ART, several studies have demonstrated that large proportions of patients experience incomplete immune restoration following treatment

**eLife digest** The human immunodeficiency virus (HIV) remains an ongoing global pandemic. There is currently no cure for HIV, but antiretroviral therapies can keep the virus in check and allow individuals with HIV to live longer, healthier lives. These drugs work in two ways. They block the ability of the virus to multiply and they allow numbers of an important type of infection-fighting cell called CD4+ T cells to rebound.

As more patients with HIV survive and transition from one life stage to the next, it is critical to understand how long-term antiretroviral therapies will affect normal age-related changes in their immune systems. The health of an immune system can be evaluated by looking at the number of CD4+ T cells an individual has, though this will vary by age and location. Clinicians use the same metrics to assess the immune health of individuals with HIV, however, as they age, it becomes a challenge to identify if a patient's immune system recovers normally or insufficiently. Thus, learning more about age-related differences in CD4+ T cells in people living with HIV may help improve their care.

Using data from 1,616 children and 14,542 adults from South Africa, Ujeneza et al. created a simple mathematical model that can compare the immune system of person with HIV with the immune system of a similarly aged healthy individual. The model shows that among individuals with HIV receiving antiretroviral therapies, children have CD4+ T-cell numbers that are closest to the numbers seen in healthy individuals of the same age. This suggests that children may be more able to recover immune system function than adults after beginning treatment. Children also start antiretroviral therapies before their immune system has been severely damaged, while adults tend to start treatment much later when they have fewer CD4+ T cells left.

Ujeneza et al. show that the fewer CD4+ T cells a person has when they start treatment, the faster the number of these cells grows after starting treatment. This suggests that the more damaged the immune system is, the harder it works to recover. This reinforces the need to identify people infected with HIV as soon as possible through testing and to begin treatment promptly. The new model may help clinicians and policy makers develop screening and treatment protocols tailored to the specific needs of children and adults living with HIV.

initiation. In Sub-Saharan Africa, approximately 10–16% of children on ART (*Mutwa et al., 2014*) and variable proportions of adults (*Barth et al., 2011*; *Fatti et al., 2014*; *Mee et al., 2008*) fail to suppress viral load within 12 months of ART initiation. Further, among those that do suppress viral load, many are suboptimal immunological responders (35–40%), that is, they do not reach a CD4+ T-cell count greater than 500 cells/μL within 5 years (*Nakanjako, 2016*; *Swiss HIV Cohort Study et al., 2005*). Some patients demonstrate no improvement at all (*Lawn et al., 2006*).

A variety of statistical methods have been used to model the recovery of CD4+ T-cell counts after ART initiation. A prior review of sub-Saharan African studies of this type established that generalized linear mixed models and generalized estimating equations were most commonly used (*Sempa et al., 2017*). However, such models are not 'mechanistic', in the sense that they make no assumptions regarding the underlying biological processes involved in CD4+ T-cell reconstitution. For this reason, these methods do not allow inferences regarding the dynamics of CD4+ recovery. Recent studies have employed an asymptotic 'semi-mechanistic' mixed model to describe CD4+ count recovery in children (*Lewis et al., 2012*; *ARROW Trial Team et al., 2013*; *Lewis et al., 2017*; *De Beaudrap et al., 2008*) and adults (*ANRS 1215/90 Study Group et al., 2009*; *Means et al., 2016*). These models simply assume that CD4+ T-cell counts follow an asymptotic recovery process in all patients. However, it has been demonstrated that in patients with suboptimal or no immune recovery, response trajectories do not follow this profile (*ARROW Trial Team et al., 2013*). Complicating this picture, diverse CD4+ T-cell count variable transformations were applied, different demographic groups were studied and there is large natural variability of such cell counts across different population groups (*Schaberg et al., 1997*; *Abuye et al., 2005*; *Maini et al., 1996*). As a result, these models do not allow for comparisons across heterogeneous groups of individuals.

In this study, we present a novel mechanistic model that describes the dynamics of CD4+ T-cell count responses in HIV-infected individuals relative to those of healthy individuals. We use nonlinear

mixed modelling methods and data from large cohorts of adults and children in South Africa. We assess the generalizability of this model from children to adults. We compare it to the previously published asymptotic model and investigate the impact of demographic and clinical characteristics on long-term immune outcomes.

## Methodology

### ART patient data

De-identified longitudinal data from HIV-infected patients receiving ART in 13 South African cohorts was provided by the International Epidemiologic Databases to Evaluate AIDS in Southern Africa (IeDEA-SA) (https://www.iedea-sa.org/) (*ART-LINC Collaboration of IeDEA et al., 2008*; *IeDEA and COHERE Cohort Collaborations and Anderegg, 2018*). The initial data set was composed of 2,858,743 CD4+ count observations for 223,688 patients of whom 202,108 were adults, 21,267 were children and 313 had no recorded date of birth or date of ART initiation. Children were defined as those aged 17 or younger at treatment initiation. After removing observations that were unrealistic (outside plausible biological ranges) or missing date of measurement, 189,647 adults and 19,060 children remained (*Figure 1*).

### 'Ratio' model construction

The ratio model relates the CD4+ T-cell dynamics from HIV+ individuals to those of healthy individuals. We first defined, then merged, separate models for each group, using the logistic growth model. This model assumes that a given quantity grows exponentially until it approaches a constant carrying capacity $k$ (*Figure 2*), or limit, to represent the dynamics of the immune system. This concept has been used in prior studies to describe the proliferation rate of CD4+ T-cells for individuals on therapy (*Kaufmann et al., 2001*; *Di Mascio et al., 2006*). We denote the CD4+ T-cell count 'growth' (or *regeneration*) rate of an individual on ART by $r$, and the 'environmental' (or *physiological*) carrying capacity by $k$, which can be thought of as the maximal number of CD4+ T-cells that can be sustained by all available biological resources.

For HIV-infected people, the rate of change of CD4+ T-cells $x$, per $\mu L$ per unit time, $t$, is expressed by the following ordinary differential equation:

$$\frac{dx}{dt} = rx\left(1 - \frac{x}{k}\right), \tag{1}$$

where $t$ starts at ART initiation. The theoretical growth rate $r$ is actually an effective rate, a constant obtained by taking the average of the growth rate at different time points. This 'instant' growth rate increases for values of $x$ smaller than the inflexion point $k/2$ and starts decreasing thereafter, as the value of $x$ approaches the carrying capacity $k$.

The solution to *Equation 1* gives the expression for CD4+ T-cell counts per $\mu L$ at time $t$ as:

$$x(t) = k\, e^{rt}\, \frac{1}{\frac{k-x_0}{x_0} + e^{rt}}, \tag{2}$$

where $x_0$ is the CD4+ T-cell count of the individual, at ART initiation.

A study on healthy South African population showed that their CD4+ counts increased from young adulthood until about 70 years of age (*Malaza et al., 2013*). Thus, we can also describe the CD4+ T-cell counts in a healthy individual $y(t)$ at time $t$, as:

$$y(t) = q\, e^{st}\, \frac{1}{\frac{q-y_0}{y_0} + e^{st}}, \tag{3}$$

where $s$ is the CD4+ T-cell growth rate, $q$ the *carrying capacity*, and time $t$ has an origin equivalent to ART initiation in an age-matched HIV infected patient. Thus, the equation describes the immune system dynamics of the healthy individual, starting from when they have a similar age as their corresponding HIV-infected counterpart.

The 'Ratio' model is then defined by dividing *Equation 2* by *Equation 3*, substituting $z_0 = \frac{x_0}{y_0}$, $K = \frac{k}{x_0}$, and $Q = \frac{q}{y_0}$, and simplifying to obtain, $z(t)$, the scaled CD4+ T-cell count at time $t$:

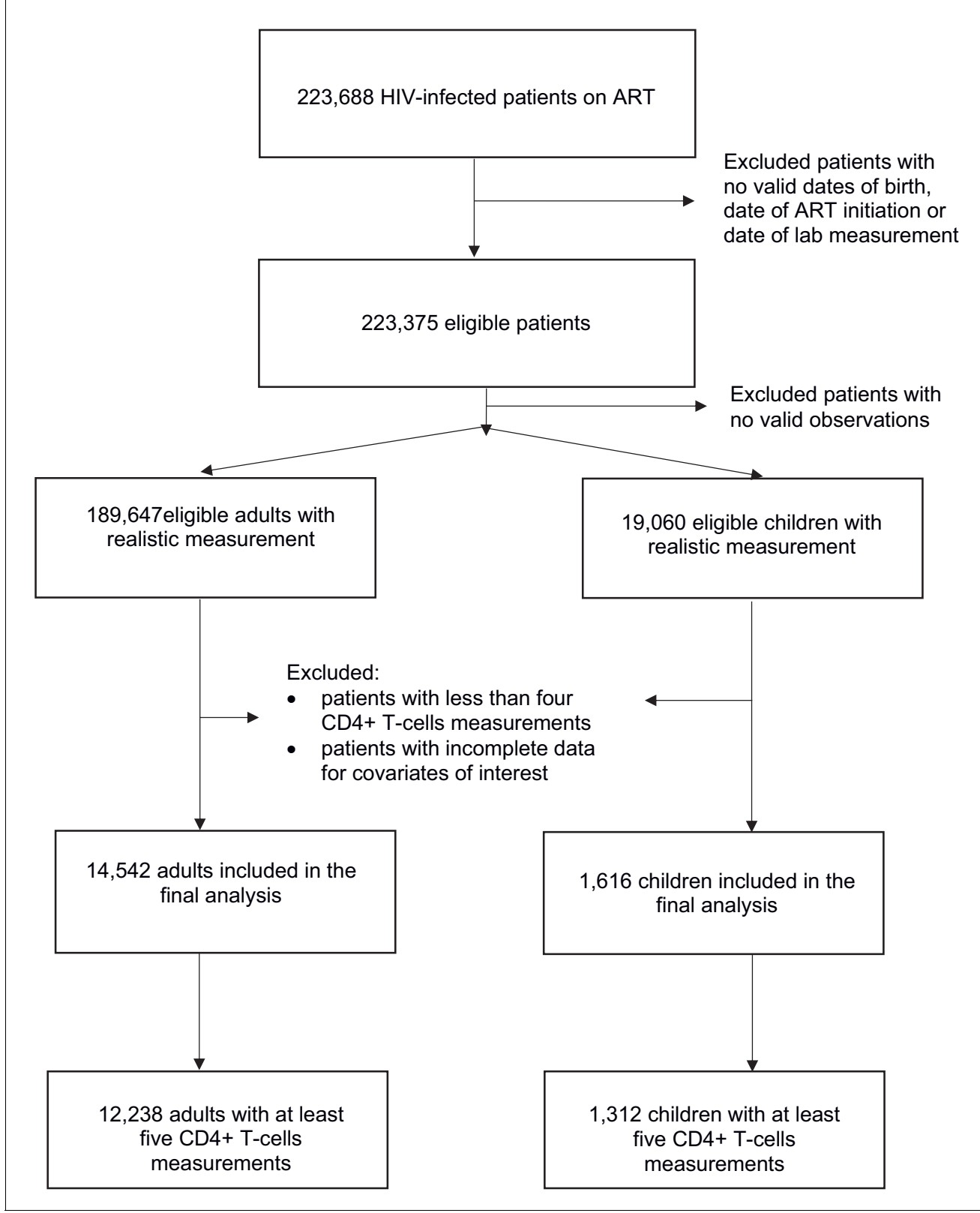

**Figure 1.** Data chart explaining the exclusion and inclusion criteria.

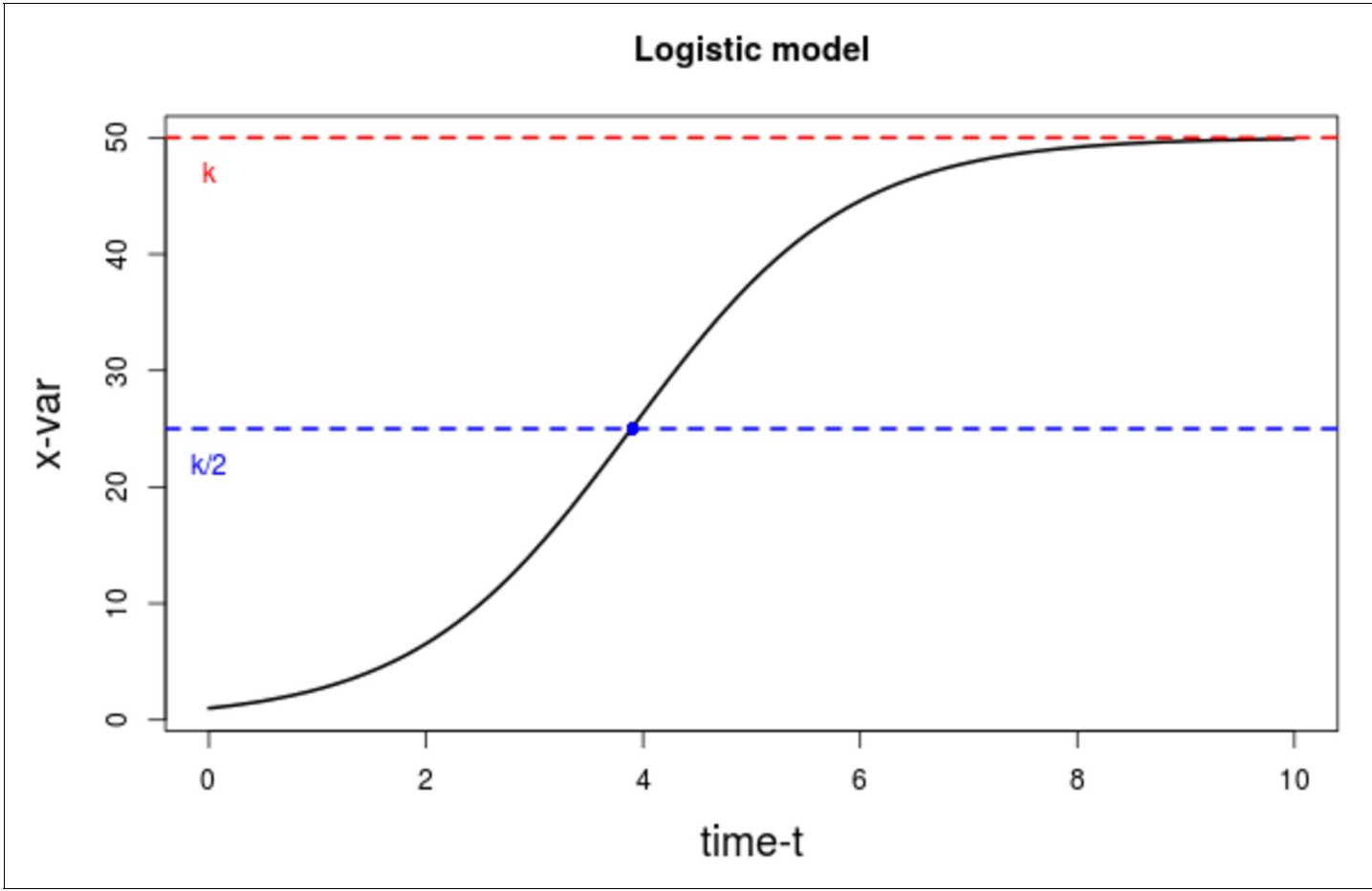

**Figure 2.** Plot of the logistic growth model. The dotted red line represents that carrying capacity k, while the dotted blue line is at the inflexion point k/2.

The online version of this article includes the following source data for figure 2:

**Source data 1.** Data source to reproduce the plot for the logistic growth model.

$$z(t) = z_0 \frac{K}{Q} \frac{1 + e^{-st}(Q-1)}{1 + e^{-rt}(K-1)}, \tag{4}$$

where $K$ and $Q$ are the CD4+ T-cell *scaled carrying capacities* of an HIV-infected and a healthy age-matched individual, respectively; and $z_0$ the ratio of the baseline CD4+ T-cell count of the HIV-infected individuals to that of the healthy individual, in order words, the *baseline scaled* CD4+ T-cell counts. An implication of this model is that when the value of $K$ is close to unity, the patient's long-term carrying capacity is approaching the particular patient's 'homeostatic optimum' for CD4+ T-cell counts, that is, the amount of CD4 cells that can be sustained by the available biological resources. In this case a subsequent increase in CD4+ T-cell counts would be unlikely. Further, associated with an 'immune set-point' achieved in healthy adults their value of $Q$ would tend to unity. In healthy children, this is not so owing to growth, that is, their CD4+ T-cell count changes with the increase of their total blood volume (*Bains et al., 2009*). Thus, their value of $Q$ is expected to be less than one.

Additionally, a *baseline scaled* CD4+ T-cell count of 1 means that the individual started therapy with a *normal* CD4+ T-cell count for their age, while a value smaller than one indicates that an individual started therapy with a lower CD4+ count compared with those of healthy individuals of the same age. Note that both growth rates for HIV-infected and healthy individuals are the same (unchanged), from the original logistic growth models to the 'ratio' model. A table indicating the ranges for each parameter is available in *Supplementary file 1*.

## Model fitting techniques

We used nonlinear mixed effects techniques (NLMM) to fit all models to data. NLMMs are characterized by two main components: a fixed effect part that describes the population mean, and the random effects that describe individual subjects' deviations from the mean. If we denote by $\theta$ the vector of population estimates, and by $b_i$ the matrix of the random effects' estimates, a general mathematical description of NLMM for continuous variables is given by,

$$z_{i,j} = f(p_{i,j}, \theta, b_i) + g(p_{i,j}, \theta, b_i, \gamma)\epsilon_{i,j}, 1 \leq i \leq N, 1 \leq j \leq n_i, \tag{5}$$

where $z_{i,j}$ is the scaled CD4+ T-cell counts of patient $i$ at time $j$; function $f$ is the nonlinear model of interest and $g$ is the structure of the error model; $N$ is the total number of subjects and $n_i$ the number of observations $j$ for each subject $i$. Both $f$ and $g$ depend on the predictor variables $p_{i,j}$, the population parameters $\theta$, and the random effects $b_i$. In addition, the function $g$ depends on the form chosen for the error model, which is governed by a set of parameters $\gamma$. In this study, we used a proportional error model, such that $g = df$ and $\gamma = d$ (single constant parameter $d$), and *Equation 4* as our basis function $f$:

$$z_{i,j} = z_{0_i} \frac{K_i}{Q_i} \frac{1 + (Q_i - 1)e^{-s_i t_{i,j}}}{1 + (K_i - 1)e^{-r_i t_{i,j}}} \times (1 + d\epsilon_{i,j}), 1 \leq i \leq N, 0 \leq j \leq n_i. \tag{6}$$

For purposes of comparison, we also applied the same NLMM methodology to an 'asymptotic' model, similar to that previously described (*Lewis et al., 2012*), such that,

$$z_{i,j} = (\text{Asym}_i - (\text{Asym}_i - \text{R0}_i)e^{c_i t_{i,j}}) \times (1 + d\varepsilon_{i,j}), \quad 1 \leq i \leq N, 0 \leq j \leq n_i, \tag{7}$$

where $\text{Asym}_i$ is the value of the asymptote for patient $i$, $\text{R0}_i$ their intercept, and $c_i$ their logarithm of rate of increase of scaled CD4+ T-cell counts. The term $(\text{Asym}_i - \text{R0}_i)$ represents the scaled increase of CD4+ T-cells following ART initiation.

The NLMM method assumes that the random effects are normally distributed, with mean zero and variance-covariance matrix $\Omega$: $b_i \sim \aleph(0, \Omega)$; and the errors $\varepsilon_{i,j}$ are also normally distributed with mean zero and variance 1: $\varepsilon_{i,j} \sim \aleph(0, 1)$. These represent noise and errors in the data. The individual parameters (denoted by a matrix $\psi_i$) are given by $\psi_i = \theta C_i + b_i$, with $C_i$ the matrix of individual covariates. A demo code is available through a github repository (https://github.com/EvaLiliane/RM_Code_eLife [copy archived at swh:1:rev:624ff31c5fc969885f29b7291ee06886d24c64f7]; *Ujeneza, 2020*).

## Variable scaling

The key outcome variable is scaled CD4+ T-cell count. In both infected adults and children, the cell counts post-ART initiation were scaled by reference values from healthy populations, to obtain the outcome variable.

For HIV-infected children these reference values were calculated from the cross-sectional data (see description in Appendix 1) of healthy children at specific ages, due to the large variability in CD4+ T-cell counts in the early years of life. For the reference values by age, a single exponential model was fitted to the healthy children's cross-sectional data and continuous population estimates were simulated (see *Appendix 1—table 1* and *Appendix 1—Figure 1*). These were within the normal CD4+ T-cell counts ranges published in South Africa (*Lawrie et al., 2009*; *Lawrie et al., 2015*) and elsewhere (*Idigbe et al., 2010*; *Pediatric AIDS Clinical Trials Group et al., 2003*). We then scaled all HIV-infected children's CD4+ T-cell counts as follows:

$$z_{i,j}(a) = \frac{x_{i,j}(a)}{y(a)}, \quad 1 \leq i \leq N, 0 \leq j \leq n_i, 0 \leq a \leq 203, \tag{8}$$

where $z_{i,j}$ is the scaled CD4+ T-cell counts of patient $i$ of age $a$ (in months) at time $j$ (measured as time since ART initiation); $x_{i,j}$ is the CD4+ T-cell counts of an HIV-infected child $i$ of age $a$ and $y(a)$ is the CD4+ T-cell counts of a healthy child of similar age $a$ as patient $i$. Scaling CD4+ T-cell counts of HIV-infected children by that of healthy children of similar ages enabled the comparison of CD4+ T-cell counts responses across ages, while simultaneously accounting for the child's growth and immune system development.

**Table 1.** Patient demographics.

| Category | Variable, *unit* | Category | Full data set | Sample for *scenario 1* |
|---|---|---|---|---|
| Children | | | | |
| Demographic | Number of patients | All | 19,060 | 1312 |
| | Median age, *years* (IQR) | All | 4.4 (1.1,8.8) | 4.5 (1.4,7.9) |
| | Baseline WHO stage (% relative to all) | Stage I | 693 (3.6%) | 31 (2.3%) |
| | | Stage II | 1232 (6.4%) | 108 (8.2%) |
| | | Stage III | 3968 (20.8%) | 614 (46.7%) |
| | | Stage IV | 2756 (14.4%) | 431 (32.8%) |
| | Median BMI z-scores | All | −0.85 (-2.2,0.2) | −0.76 (-1.98,0.26) |
| | Gender | Female | 9606 (50.4%) | 674 (51.3%) |
| | | Male | 9454 (49.6%) | 638 (48.6%) |
| | Median time on ART, *years* (IQR) | All | 2.0 (0.0,4.0) | 4.0 (3.0,5.0) |
| | Year of ART initiation (IQR) | | 2004–2012 | 2004–2012 |
| Clinical characteristics | CD4+ T-cell count at baseline, *count/*μL (IQR) | | 493 (229,890) | 404 (159,706) |
| | Median scaled CD4+ T-cell count at baseline, (IQR) | | 0.30 (0.15,1.50) | 0.24 (0.09,0.42) |
| | Median viral load at baseline, per 1000 copies/mL (IQR) | | 150 (20,7342) | 155 (29,670) |
| | Median log viral load at baseline, copies/mL (IQR) | | 5.1 (4.3,6.8) | 5.1 (4.4,5.8) |
| | Number of patients that suppressed viral load within 12 months of treatment initiation (% relative to non-missing) | Yes | 1673 (26%) | 479 (36.5%) |
| | | No | 4764 (74%) | 833 (63.5%) |
| Adults | | | | |
| Demographic | Number of patients | All | 189,647 | 12,238 |
| | Median age, *years* (IQR) | All | 35 (29,42) | 36 (30.6,42.9) |
| | Baseline WHO stage (% relative to all) | Stage I | 20,711 (10.9%) | 1588 (12.9%) |
| | | Stage II | 15,815 (8.3%) | 817 (6.6%) |
| | | Stage III | 42,393 (22.3%) | 4050 (33%) |
| | | Stage IV | 11,466 (6%) | 1336 (10.9%) |
| | Gender | Female | 124,006 (65.4%) | 6944 (56.7%) |
| | | Male | 65,641 (34.6%) | 5294 (43.2%) |
| | Median time on ART, *years* (IQR) | All | 1.0 (0.0,3.0) | 3.0 (2.0,5.0) |
| | Year of ART Initiation *IQR* | | 2004–2012 | 2004–2012 |

*Table 1 continued on next page*

*Table 1 continued*

| Category | Variable, *unit* | Category | Full data set | Sample for *scenario 1* |
|---|---|---|---|---|
| Clinical characteristics | Median CD4+ T-cell count at baseline, count/$\mu L$ (IQR) | | 140 (67,206) | 128 (62,196) |
| | Median scaled CD4+ T-cell count at baseline (IQR) | | 0.18 (0.08,0.26) | 0.16 (0.07,0.24) |
| | Median viral load at baseline, per 1000 copies/mL (IQR) | | 27 (0.8,132.3) | 39 (2.3, 151) |
| | Median log viral load at baseline, copies/mL (IQR) | | 4.4 (2.9,5.1) | 4.5 (3.3,5.1) |
| | Number of patients that suppressed viral load within 12 months of treatment initiation (% relative to relative non missing) | Yes | 10,746 (36.2%) | 5011 (40.9%) |
| | | No | 18,945 (63.8%) | 7227 (59.0%) |

Note: A z-score BMI of −1 indicates that the child's body mass index is at one standard deviation below the body mass of a healthy child, while a z-score BMI of 0 means that the child has normal body mass for his/her age.

For adults, normal reference values were estimated using values obtained from the literature (*Malaza et al., 2013*; *Lawrie et al., 2009*, *Ngowi et al., 2009*; *Lugada et al., 2004*; *Institute of Human Virology/Plateau State Specialist Hospital AIDS Prevention in Nigeria Study Team et al., 2005*; *Oladepo et al., 2009*) (see details in *Appendix 1—table 2* and *Appendix 1—figure 2*). Although healthy adult CD4+ T-cell counts are known to vary with age, this variability is small compared to that observed in children, with an average difference of 100 cells between 25 and 60 years old healthy adults [*Oladepo et al., 2009*] vs 1500 cells difference between 3 months and 15 years old healthy children [*Lawrie et al., 2015*]. Similarly, average CD4+ T-cell counts differences between male and female adults in South Africa were in the range of 80–150 cells (*Malaza et al., 2013*; *Lawrie et al., 2009*). We evaluated results obtained from scaling HIV-infected CD4+ counts by simulated age and sex-dependent reference CD4+ counts values, and found minor or no difference with those obtained from CD4+ T-cell counts scaled by a single value. We obtained exactly the same population estimates and very similar individual parameter estimates. These were evaluated and we found no significant differences (see *Supplementary file 2*). Thus, due to the unavailability of individual age-specific data for a South African healthy adult population, a single *normal* reference value, that is, a constant $y$ of 800 CD4+ T-cells per µL, was used to scale all CD4+ T-cell count observations for adults on ART ($z_i = \frac{x_{i,j}}{y}$).

## Baseline scaled CD4+ T-cell counts

Our analysis was restricted to patients with available baseline (i.e. at antiretroviral treatment initiation) CD4+ T-cell counts and sufficient observations to estimate all model parameters (*Figure 1*). We categorized as baseline any CD4+ count measurement that was taken within 15 days before or after the ART initiation date. Two model-fitting scenarios were defined based on the availability of enough observations to estimate the model parameters:

**Table 2.** BIC comparison of the unadjusted ratio and asymptotic models, under different scenarios for baseline scaled CD4+ T-cell counts $z_0$.

| | | Scenario 1: $z_0$ estimated | | Scenario 2: $z_0$ as a predictor | |
|---|---|---|---|---|---|
| | | Ratio model | Asymptotic model | Ratio model | Asymptotic model |
| Adults | Sample size | 12,238 | | 14,542 | |
| | BIC | −134,016.7 | −126,716.2 | −178,702.3 | −186,244.2 |
| Children | Sample size | 1312 | | 1616 | |
| | BIC | −2,137.1 | −1,523.1 | −6,028.8 | −7,969.7 |

Scenario 1: In which baseline scaled CD4+ T-cell counts were estimated, including 1312 children and 12,238 adults, with a minimum of five CD4+ T-cell counts measurements and no missing values for our variables of interest (specified below).

Scenario 2: Where baseline scaled CD4+ T-cell counts were used as a predictor, including 1616 children and 14,542 adults, with a minimum of four CD4+ T-cell counts measurements and no missing values for our variables of interest. These variables were CD4+ T-cell counts since ART initiation; viral load, age and body mass index at ART initiation; sex and suppression (or not) of viral load within 12 months of ART initiation.

In this paper we will mostly discuss results of scenario 1, where all parameters are estimated. The methodology used and most of the obtained results are applicable for scenario 2, when baseline CD4+ T-cells counts data are available. We also present results of scenario 2.

## Structure of the random effects

Under both scenarios, in the initial model fitting, all parameters were assumed to vary per individual (i.e. there were random effects on all parameters) and their distributions was set as log-normal. Thus, for each parameter $\theta$, the distribution of individual values was $log(\theta_i) \sim \aleph(\mu, \omega^2)$, with mean $\mu$ and variance $\omega^2$. Random effects were assumed to be independent and the identity diagonal matrix was used for the variance–covariance structure.

This assumption was later relaxed and different variance–covariance structures were subsequently evaluated. To compare models, the Akaike and Schwarz information criteria (AIC and BIC respectively), computed by importance sampling, were used (see *Comets et al., 2017*). Models with a full variance–covariance matrix were retained as this structure gave the lowest AIC and BIC values.

## Covariates

Our choice of covariates was based on literature reviews (*Pinzone et al., 2012*; *Sempa et al., 2017*), data availability, and biological plausibility. We considered baseline characteristics that were measured within 15 days before or after ART initiation, namely age, z-score body-mass-index (BMI, for children), and viral load. Other covariates included sex and viral suppression (or not) within 12 months of ART initiation.

Patient age at ART initiation was expressed in months. Z-score BMI at baseline for children was calculated using WHO-Igrowup's package (*WHO Multicentre Growth Reference Study Group,, 2006*). 'Suppress', a binary, was defined as reaching an undetectable viral load (<1000 copies/mL) or not, within 12 months of ART initiation. Baseline viral load was log transformed to simplify model fitting. Baseline age and sex effects were included for all parameters, while viral load suppression, baseline log viral load and BMI z-scores for children, were only included for the parameters describing the longitudinal immune responses of individuals on ART, namely: *scaled carrying capacity* post-ART, CD4+ T-cell *growth rate* post-ART and *baseline scaled CD4+ T-cell count* .

## Additional considerations

Final adjusted models were obtained using a backward–forward stepwise approach using p-value criterion for covariate inclusion and AIC, BIC for the overall models. To evaluate the robustness of our results, we compared the estimated parameters of the full analysis described above using only three covariates: sex, baseline age, and baseline viral load, with those from models with all covariates.

All graphs and computations were produced in the statistical environment R (*R Development Core Team, 2017*). Nonlinear mixed model fitting employed the saemix R package (*Comets et al., 2005*), which uses a stochastic approximation, expectation-maximization algorithm for parameter estimation. All final adult and children models (*Figures 3* and *4*) converged relatively rapidly towards their estimated values, as shown by the log-likelihood graphs (*Figure 3—figure supplement 1* and *Figure 4—figure supplement 1*). Even when initial values were marginally varied, the final models converged to similar estimates.

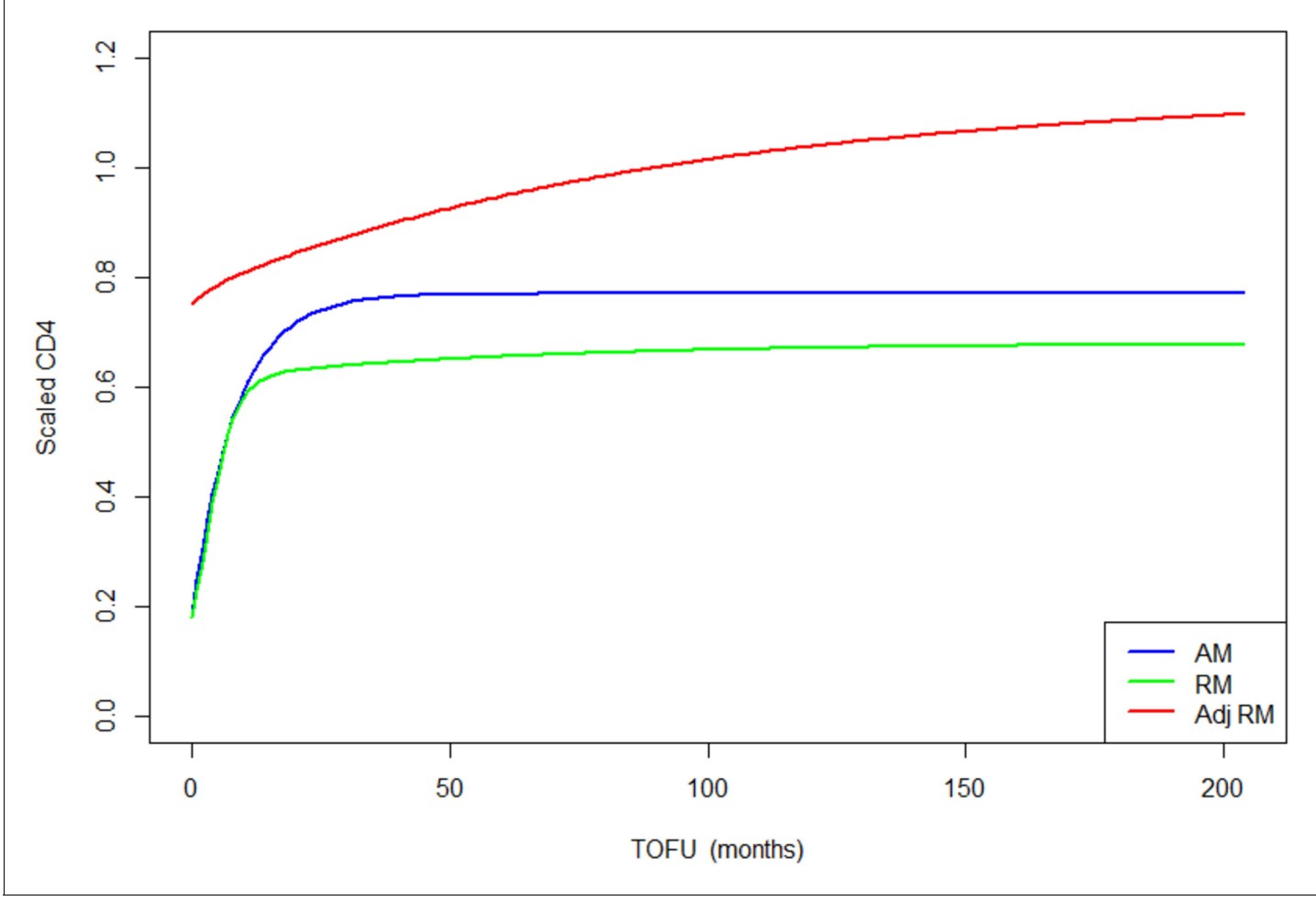

**Figure 3.** Children population-level CD4 trajectory, as estimated by the unadjusted ratio and asymptotic models, and the adjusted ratio model. Simulation of population-level CD4+ count trajectory for children, from unadjusted fixed estimates of the asymptotic model (AM) in blue and the ratio model (RM) in green. The red line represents simulation from the adjusted population-level RM estimates. Parameters used for the AM are presented in *Supplementary file 3 – Table 1*, scenario 1. Those used for the RM are estimated fixed effect for the null model (not shown in the paper): K = 3.4, Q = 0.9, r = 0.35, s = 0.017, z0 = 0.18. Fixed effect presented in *Table 3* (scenario 1) are used for the adjusted ratio model (Adj RM), for baseline covariates z-score BMI, age, log viral load; and sex and suppression of viral load within 12 months of starting therapy. Convergence plots for the Adj RM are given in *Figure 3—figure supplement 1*, and simulation of individual fits in *Figure 3—figure supplement 2*.

The online version of this article includes the following source data and figure supplement(s) for figure 3:

**Source data 1.** Data source to reproduce the population-level CD4 trajectories plot for children.

**Figure supplement 1.** Convergence plots for children adjusted ratio model.

**Figure supplement 2.** Sample of individual children plots for the adjusted ratio model.

**Figure supplement 2—source data 1.** Data source to reproduce the individual-level CD4 trajectories plot for children.

## Results

### Clinical characteristics

Cohort characteristics are summarized in *Table 1*. For scenario 1, the median number of clinical visits was 8, per patient, in both adults IQR (6,9) and children IQR (6,10), with a median follow-up time of 3 years for adults IQR (2,5) and 4 years for children IQR (3,5). At baseline, that is, ART initiation, the median age for children was 4.5 years IQR (1.4, 7.9) and 36 years IQR (30.7, 43) for adults. A clinical WHO stage III was most common in all patients at ART initiation. Among adults, there were more females than males (56.7% versus 43.2% respectively), but only slightly more females among children (51.3% versus 48.6% males). All patients in our data sets initiated therapy between 1997 and 2013, with 95% of them initiating in the period, 2003–2012. About 53.4% of children reached a CD4+

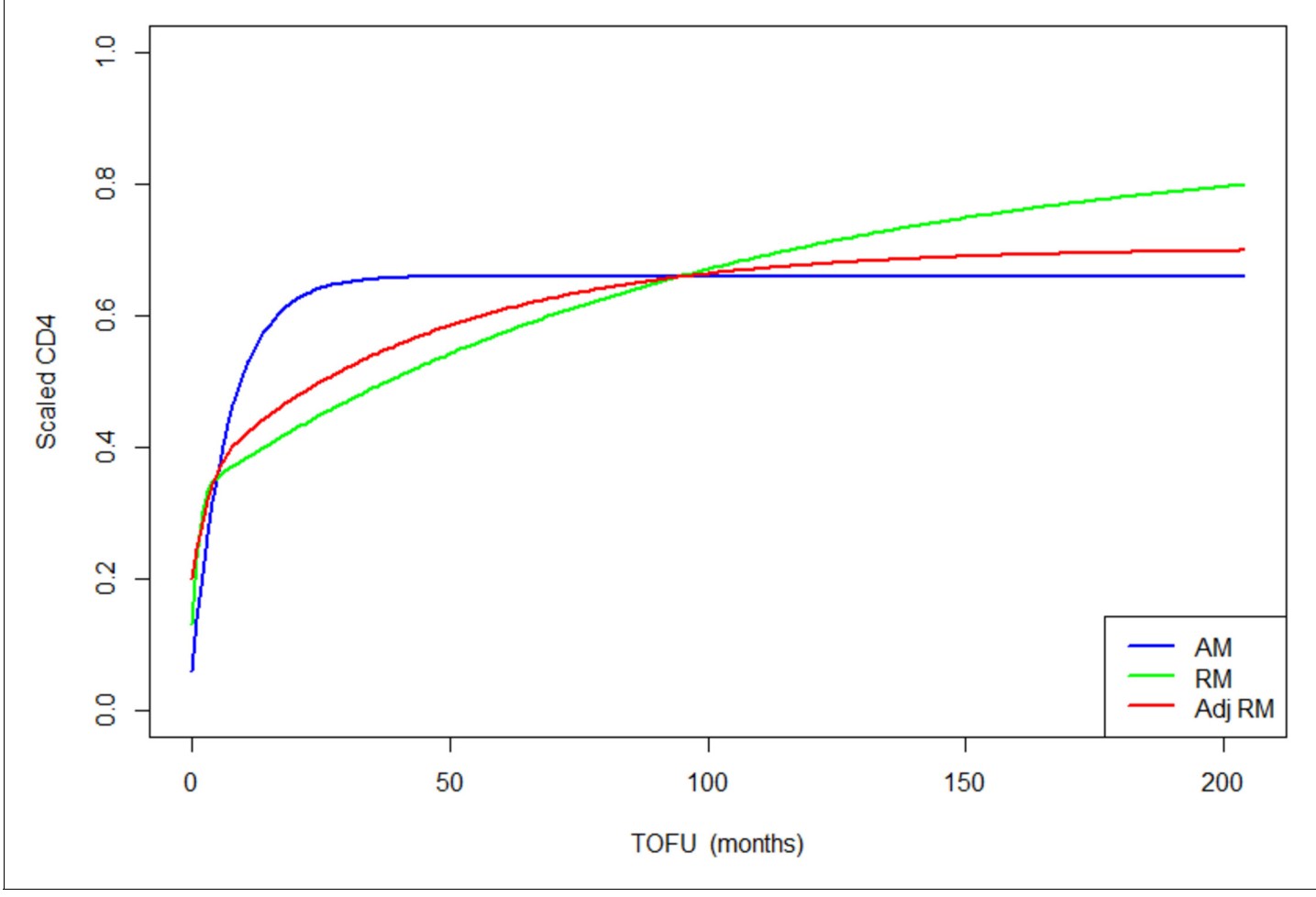

**Figure 4.** Adults population-level CD4 trajectory, as estimated by the unadjusted ratio and asymptotic models, and the adjusted ratio model. Simulation of population-level CD4+ count trajectory for adults, from unadjusted fixed estimates of the asymptotic model (AM) in blue, and the ratio model (RM) in green. The red line represents simulation from the adjusted population-level RM estimates. Parameters used for the AM are presented in *Supplementary file 3 – Table 1*, scenario 1. Those used for the RM are estimated fixed effect for the null model (not shown in the paper): K = 2.54, Q = 0.38, r = 1.23, s = 0.01, z0 = 0.13. Fixed effect presented in *Table 3* (scenario 1) are used for the adjusted ratio model (Adj RM), for baseline covariates sex, age, log viral load, and suppression of viral load within 12 months of starting therapy. Convergence plots for the Adj RM are given in *Figure 4—figure supplement 1*, and simulation of individual fits in *Figure 4—figure supplement 2*.

The online version of this article includes the following source data and figure supplement(s) for figure 4:

**Source data 1.** Data source to reproduce the population-level CD4 trajectories plot for adults.

**Figure supplement 1.** Convergence plots for adults adjusted ratio model.

**Figure supplement 2.** Sample of individual adult plots for the adjusted ratio model.

**Figure supplement 2—source data 1.** Data source to reproduce the individual—level CD4 trajectories plot for adults.

T-cell count of 500 per μL within 12 months of ART initiation, while only 4.4% of adults reached the same threshold. This number increases to 10.1% after 2 years of therapy for adults.

The median baseline CD4+ T-cell counts for children and adults were low at 404 per μL IQR (159.7,706.2) and 128 per μL IQR (62,196) respectively (*Table 1*). Higher counts at ART initiation were more common in younger (0–5 year's age group) versus older children (>5 years). In adults, no differences were found in CD4+ T-cell counts at baseline, for different age groups. Both children and adults that presented with higher counts at baseline had lower viral loads versus those with low baseline counts.

## Comparison of models and random effects structures

The performance of both asymptotic and ratio models, in terms of AIC and BIC criteria, was found to be strongly dependent on the structure selected for the variance–covariance matrices (see *Supplementary file 3*). Initially, both models were defined with random effects on all parameters and a diagonal matrix structure for the random effects, that is, individual random effects were assumed independent. Different matrix structures were then compared and those with full matrices, that is, in which random effects are correlated, were found to be the best.

For both adults and children, the AIC and BIC for the Ratio model were smaller than for the Asymptotic model when *baseline scaled* CD4+ T-cell counts were estimated (*Scenario 1*). When the *baseline scaled* CD4+ T-cell counts were used as a predictor (*Scenario 2*) this reversed. However, the number of parameters estimated by the two models was different. For the Asymptotic model there were three for scenario one vs two for scenario 2, a 33% change, and for the ratio model there five for scenario one vs four for scenario 2, a 20% change (*Table 2*). Given the different numbers of parameters, the Ratio performed well in comparison to the Asymptotic model.

In terms of parameter estimations, both models estimated baseline scaled CD4+ of similar magnitudes, for HIV-infected adults and children (*Table 3* and *Supplementary file 4 – Tables 1* and *2*). The asymptotic model predicts a sharp increase of CD4+ counts, which rapidly reaches an asymptote, while the ratio model predicts a sharp increase that is followed by a slower but still gradual increase of scaled CD4+, up to more than 10 years after ART initiation (*Figures 3* and *4*).

We note that the parameter estimates obtained using a baseline CD4 scaling constant of 800 cells/μL for healthy adults were comparable to those obtained using age-dependent healthy adults CD4 values (*Supplementary file 2*). Similarly, the estimates obtained using a baseline CD4 scaling constant of 800 cells/μl for healthy adults were also comparable to those obtained without scaling (i.e. using a scaling constant of 1 cell/μL) for all parameters except $z_0$, whose estimate increased by about 800-fold (*Supplementary file 2*) as expected.

## Final ratio models

We found no major differences in the estimated fixed effects when adjusting the models for age and sex only, versus adjusting for all our covariates. The minor differences between both fits were the sizes of estimated parameters (*Supplementary file 4 – Table 2*), a few inconsistences in the effects of age on particular parameters (adults and children) and sex effects on post-ART scaled carrying capacity (adults only). Therefore, we only describe results for the models adjusted for all covariates. These results are presented in *Table 3*. Note that a sample of individual fits is shown in *Figure 3— figure supplement 2* and *Figure 4—figure supplement 2*.

### Population estimates/fixed effects

Both adults and children, on average, start ART with an impaired immune system. We found that baseline scaled CD4+ T-cell count was 0.20 (95% CI: 0.18, 0.22) for adults and 0.74 (95% CI: 0.49, 1.00) for children (Scenario 1, *Table 3*).

The *growth rates* of CD4+ T-cell counts were more than tenfold higher in HIV infected versus healthy individuals. We found that in adults, the growth rate for HIV-infected individuals was 0.49 cell counts per μL per day, while it was 0.02 for healthy individuals. In infected children, a growth rate of 0.27 cell counts per μL per day was found, while it was much lower (0.01) in healthy children. This means, for example, that an infected child starting ART with 100 cells will cross the 500 cells threshold in 1470 days, whereas an infected adult with the same number of cells at ART initiation will cross the threshold much faster, after about 816 days.

For HIV-infected individuals the *scaled carrying capacity* was higher than that of healthy individuals in both adults and children. We found that the scaled carrying capacity for HIV-infected adults was 1.75, which is three times higher than that of healthy adults (0.49). We found that the scaled carrying capacity for HIV-infected children was above one (1.03). The scaled carrying capacity post-ART of 1.75 in adults implied that an average 36.5 year old adult on ART would have to experience a 75% increase of their initial CD4+ T-cells to reach their ultimate CD4+ T-cell counts levels. For an average 5-year-old child, this rebound needs to be much lower: 3% increase of their initial CD4+ T-cell counts, to reach normal ranges.

**Table 3.** Ratio model estimated parameters for children and adults.
*** means significant at 99%, and ** significant at 95%.

| | Children | | Adults | |
|---|---|---|---|---|
| **Model** | **Scenario 1:** $z_0$ **estimated (1312 subjects) BIC = −2,457.019** | **Scenario 2:** $z_0$ **as a predictor (1616 subjects) BIC = −6,156.762** | **Scenario 1:** $z_0$ **estimated (12,238 subjects) BIC = −136,284.7** | **Scenario 2:** $z_0$ **as a predictor (14,542 subjects) BIC = −180,557.8** |
| **Variable** | **Estimate (95% CI)** | **Estimate (95% CI)** | **Estimate (95% CI)** | **Estimate (95% CI)** |
| *Scaled carrying capacity post ART (K)* | 1.03 (0.73,1.34) | 1.13 (0.84, 1.41) | 1.75 (1.59, 1.90) | 1.69 (1.57, 1.80) |
| Sex, *ref is male* | - | - | −0.06 (−0.09,−0.03)*** | - |
| Age, *month* | 0.005 (0.004, 0.007)*** | 0.0039 (0.002, 0.004)*** | −0.0007 (−**0.008**,−**0.0006**)*** | −0.0006 (−0.0007,−0.0004)*** |
| BMI | −0.12 (−0.15,−0.08)*** | −0.09 (−0.12,−0.06)*** | - | - |
| Log viral load | 0.06 (0.04, 0.08)*** | 0.07 (0.05, 0.09)*** | 0.069 (0.065, 0.073)*** | 0.061 (0.058, 0.065)*** |
| Suppress, *ref is no* | - | 0.001 (−0.096, 0.092)** | 0.11 (0.08, 0.14)*** | 0.12 (0.10, 0.15)*** |
| *Scaled carrying capacity healthy individuals (Q)* | 0.68 (0.57, 0.78) | 2.02 (1.59, 2.44) | 0.49 (0.47, 0.52) | 0.46 (0.36,0.56) |
| Sex, *ref is male* | - | - | −0.55 (−0.63,−0.47)*** | −0.54 (−0.63,−0.46)*** |
| Age, *month* | −0.005 (0.003, 0.007)*** | - | - | 0.0002 (−0.0001, 0.0006) |
| *Scaled CD4+ T-cell count at ART initiation ($z_0$)* | 0.74 (0.49, 1.00) | - | 0.20 (0.18, 0.22) | - |
| Sex, *ref is male* | - | - | 0.08 (0.05, 0.12)*** | - |
| Age, *month* | −0.007 (−0.009,−0.006)*** | - | 0.0006 (0.0004,0.0008)*** | - |
| BMI | 0.15 (0.11, 0.19)*** | - | - | - |
| Log viral load | −0.070 (−0.096,−0.044)*** | - | −0.081 (−0.087,−0.076)*** | - |
| *Rate of growth of CD4+ healthy individuals, cells per µL per day (s)* | 0.011 (0.007, 0.016) | 0.002 (0.001, 0.003) | 0.022 (0.016, 0.028) | 0.02 (0.01,0.03) |
| Sex, *ref is male* | - | - | −0.40 (−0.53,−0.28)*** | −0.50 (−0.62,−0.37)*** |
| Age, *month* | 0.008 (0.004, 0.017)*** | 0.015 (0.010, 0.020)*** | −0.0012 (−0.0017,−0.0007)*** | −0.0016 (−0.0022,−0.0010)*** |
| *Rate of growth of CD4+ post ART, cells per µL per day (r)* | 0.27 (0.160,0.32) | 0.17 (0.12, 0.21) | 0.49 (0.39, 0.60) | 0.79 (0.60, 0.97) |
| Sex, *ref is male* | | | - | −0.19 (−0.33,−0.04)** |
| Age, *month* | −0.001 (−0.003, 0.0001)** | - | - | - |
| BMI | −0.08 (−0.12,−0.04)*** | −0.07 (−0.11,−0.03)*** | - | - |
| Log viral load | 0.03 (0.00, 0.62)** | 0.05 (0.03, 0.08)*** | 0.11 (0.09, 0.13)*** | 0.11 (0.09, 0.13)*** |
| Suppress, *ref is no* | −0.12 (−−0.27, 0.02) | −0.15 (−0.28,−0.03)*** | −0.57 (−0.70,−0.44)*** | −0.62 (−0.76,−0.47)*** |

## Individual estimates and random effects

For both adults and children, the coefficient of variation for the random effects for all parameter estimates was lower for the parameters of HIV-infected patients (*k, r,* and *z0*) and higher for the healthy individual parameters (*q* and *s*). This is understandable given the lack of individual matching in the data, that is, a person could not be HIV-infected and healthy simultaneously.

As can be seen in *Table 4*, the correlation between the random effects was similar in adults and children, with the exception of the correlation between the *scaled carrying capacity* and *growth rate* of CD4+ T-cells for healthy individuals, that was positive and strong (0.61) in adults, but weak and negative (−0.13) in children. The strongest correlation was between the *scaled carrying capacity* for

**Table 4.** Correlations of individual random effects.
Scenario 1 below the diagonal. Scenario 2 above.

| | | $K$ | $Q$ | $z_0$ | $s$ | $r$ |
|---|---|---|---|---|---|---|
| Scaled carrying capacity post ART – $K$ | Adults | Scenario 2 | 0.40 | - | 0.48 | −0.28 |
| | Children | Scenario 1 | −0.03 | - | 0.57 | 0.01 |
| Scaled carrying capacity healthy ind – $Q$ | Adults | 0.30 | Scenario 2 | - | 0.37 | −0.41 |
| | Children | 0.10 | Scenario 1 | - | −0.47 | −0.24 |
| Baseline scaled CD4+ T-cell count - $z_0$ | Adults | −0.67 | 0.27 | Scenario 2 | - | - |
| | Children | −0.85 | 0.38 | Scenario 1 | - | - |
| Rate of growth of CD4+ healthy ind – $s$ | Adults | 0.36 | 0.61 | −0.30 | Scenario 2 | 0.15 |
| | Children | 0.45 | −0.13 | −0.52 | Scenario 1 | −0.22 |
| Rate of growth of CD4+ post ART – $r$ | Adults | −0.23 | −0.33 | −0.30 | 0.23 | Scenario 2 |
| | Children | −0.03 | −0.40 | −0.25 | 0.32 | Scenario 1 |

In red: Opposite direction.
Underlined: Difference of correlation between children and adults.

HIV-infected individuals and *baseline scaled* CD4+ T-cell counts, in both adults (−0.67) and children (−0.85).

## Covariate estimates in children

We found that being 1 month older resulted in a decrease in *baseline scaled* CD4+ T-cell count of −0.007 (95% CI: −0.009, −0.006). Similarly, being 1 month older was associated with a higher CD4+ T-cell *growth rate* in healthy children (0.008) and a significant decreasing effect on post-ART CD4+ T-cell growth rate (−0.001). This means that, given a threshold of 200 CD4+ cell counts, younger children will reach it earlier than older children that had a similar CD4+ T-cell count at baseline. Furthermore, we found that age had a decreasing effect on the *scaled carrying capacity* of healthy children (−0.005).

Both scenarios show similar effects of baseline viral load on children's parameters. We found that one unit increase in baseline log viral load was associated with a −0.07 decrease in the baseline scaled CD4+ T-cell count, and a +0.06 increase in the CD4+ T-cell *scaled carrying capacity* post-ART. It also led to a +0.03 increase in the *growth rate* of CD4+ T-cells post-ART which indicates that those with higher viral loads at ART initiation will reach a defined threshold faster than those with lower viral loads, if they started with the same initial number of cells.

Baseline BMI z-score was associated with children's *baseline scaled* CD4 T-cell count, post-ART CD4+ T-cell *scaled carrying capacity*, and *growth rate*. One unit increase in baseline BMI z-score resulted in an increase of +0.15 in *baseline scaled* CD4+ T-cell count, a decrease in the *growth rate* of CD4+ T-cells post-ART (−0.08) and in the post-ART *scaled carrying capacity* (−0.12). An untreated child with advanced HIV-disease is likely to have a low BMI z-score and will consequently have a low scaled CD4+ T-cell count at ART initiation.

## Covariate estimates in adults

In adults, being 1 month older was associated with an increase of +0.0006 (95% CI: 0.0004, 0.0008) in *baseline scaled* CD4+ T-cell counts. We found no effect of age on post-ART CD4+ T-cell growth rate, but it was associated with a decline in CD4+ T-cell *growth rate* in healthy adults (−0.0012 per month older). Our results show that being older was associated with a decrease in the *scaled carrying capacity* in adults on ART (−0.0007 per month older), though it had no effect on the *scaled carrying capacity* of healthy adults.

Sex and baseline log viral load had an effect on CD4+ T-cell dynamics in adults. We found that adult females started ART therapy at higher scaled CD4+ T-cell values compared to males (+ 0.08) and one unit increase in baseline log viral load was associated with a decrease in the *baseline scaled CD4+ T-cell counts* (−0.08). We also observed a +0.07 increase in the CD4+

T-cell *scaled carrying capacity* post-ART and a +0.11 increase in the *growth rate* of CD4+ T-cells post-ART, per unit of increase in baseline log viral load.

## Discussion

CD4+ T-cell count remains the main proxy used to evaluate the long-term effects of ART on the immune system. Understanding its dynamics is crucial, to ensure that the best care is delivered to HIV-infected patients. Although previous studies described CD4+ T-cell count dynamics in HIV-treated patients, using non-mechanistic (*Sempa et al., 2017*) and mechanistic models (*Lewis et al., 2012*; *Means et al., 2016*; *Di Mascio et al., 2006*), none provided a method that allows for direct comparison with age-matched healthy controls. This study proposes a mechanistic model for the immune system reconstitution, which relates CD4+ T-cells count of an individual on ART with that of a healthy individual with similar characteristics. We demonstrate that CD4+ T-cell *growth rates* are higher in HIV-treated patients than in healthy individuals and that age has opposite effects on CD4+ counts dynamics in HIV-treated children, compared to healthy children. This model was compared to the asymptotic model, previously used to model CD4+ T-cell trajectories after ART initiation, under two different scenarios: estimating baseline scaled CD4+ T-cell count (scenario 1) and using it as a predictor (scenario 2). Moreover, this study is the first of its kind in evaluating large samples of children and adults data in a comparative way.

*Corbeau and Reynes, 2011* have described the three phases of CD4+ counts recovery in HIV-treated patients: a sharp increase in the first 1–6 months, a still high increase up to 2 years, and a slow gradual increase that goes on beyond 4 years of therapy. Our results show that though both the ratio and asymptotic model predict similar scaled baseline CD4+ counts, only the ratio model is able to reproduce these three phases of immune reconstitution. Thus, it is likely to be better in accurately predicting the long-term behaviour of CD4+ T-cell counts trajectory. Furthermore, the ratio model is derived directly from the logistic growth model which describes a known biological process, and as such, CD4+ T-cells *growth rate r* is directly relatable to the average of the actual increase of CD4+ T-cells per month over the observation period.

### Baseline scaled CD4+ T-cell counts (z0)

Fitting our model to the data demonstrated that larger values of baseline scaled CD4+ T-cell counts are associated with larger values of scaled CD4+ T-cell counts at any subsequent time. This is the consequence of *equation 4*, which is in line with the findings of prior studies in which higher baseline CD4+ T-cell counts were associated with a higher final *or plateauing* value for CD4+ T-cell counts (*Moore and Keruly, 2007*). Similarly, patients with very low baseline CD4+ T-cell counts, <350 cells/μL, often fail to reach normal values even after long durations of therapy (*Nakanjako, 2016*; *Swiss HIV Cohort Study et al., 2005*; *Moore and Keruly, 2007*; *Kelley et al., 2009*). We also found that the scaled carrying capacity of CD4+ T-cells in HIV-infected individuals was negatively correlated with the baseline scaled CD4+ T-cell counts in both adults and children, respectively, −0.64 and −0.81. This seems reasonable in that the closer an individual is to their normal or *optimal* CD4+ T-cell value at ART initiation, the less cell population expansion is required to reach normal levels.

### Scaled carrying capacity of CD4+ T-cells for HIV-infected and healthy individuals

Our parameter estimates demonstrated that the scaled cellular carrying capacity was higher in individuals on ART than in those who were healthy. In both HIV-infected adults and children the scaled carrying capacity post-ART was greater than 1, meaning that baseline CD4+ T-cell counts were lower than their corresponding long-term *homeostatic optimum*. That is, individual CD4+ T-cell counts had to grow to reach normal levels. This is consistent with the notion that an impaired immune system usually experiences repair following treatment initiation.

In contrast, the scaled capacity for healthy adults and children were both lower than unity. In children the value was 0.68, which is consistent with a mechanistic understanding of the dynamics of healthy CD4+ T-cell counts in early life. In particular, there is an increase until the age of approximately 1 year and then a decrease thereafter (*Bains et al., 2009*). Total blood and body volume increase throughout childhood associated with the shrinkage of the thymus, which is accompanied by a reduction of naïve T-cell production (*Hapuarachchi et al., 2013*; *Hazenberg et al., 2004*). In

combination, this results in decreasing CD4+ T-cell counts per volume with age. In healthy adults, we found a scaled carrying capacity of 0.49, suggesting a decreasing CD4+ T-cell trend with age. This is in agreement with other studies from elsewhere which found that CD4+ T-cell counts decreased from young adulthood to middle age (*Lugada et al., 2004*; *Zeh et al., 2011*). However, one study in healthy South African adults has described CD4+ T-cell counts increasing slightly until the age of 64 years old (*Malaza et al., 2013*).

## Post-ART CD4+ T-cell count growth rate depends on scaled baseline CD4+ T-cell counts

We found that post-ART CD4+ T-cell growth rate depends inversely on the cell count at treatment initiation, that is, they are negatively correlated (*Table 4*). This implies that the higher the scaled CD4+ T-cell counts of an individual at ART initiation, that is, the closer it is to its healthy/normal true value, the lower the rate of recovery will be. This is reasonable given a decrease in the *need* to achieve normal levels and is in agreement with the findings of prior studies (*Lawn et al., 2006*; *Sachsenberg et al., 1998*).

## Post-ART CD4+ T-cell growth rates in HIV-infected versus healthy individuals

We found that CD4+ T-cell growth rates were higher in individuals on ART than in healthy individuals. This agrees with prior studies in which cell growth rates in adults on ART were estimated to be sixfold to tenfold greater than in healthy adults (*Sachsenberg et al., 1998*; *Hazenberg et al., 2000*). This is understandable bearing in mind the immune system's effort after treatment initiation to re-fill the void of peripheral CD4+ T-cells destroyed during HIV infection. Biological studies have described this as an initial redistribution of memory T-cells from the lymph nodes into the blood stream, followed by homeostatic proliferation and production of naïve cells by the thymus (*Tsukamoto et al., 2009*; *Autran, 1999*). Thus, given that HIV-infected individuals have much lower CD4+ T-cell counts they require higher CD4+ T-cell growth rates than healthy individuals. This behaviour was captured by our model.

Interestingly, the differences in our estimates for CD4+ T-cell growth rate in healthy individuals versus patients on ART were slightly lower than those found elsewhere (*Sachsenberg et al., 1998*). This may be due to: the differences in the total time of follow-up of our patients, selection bias in the original population data, differences in the demographics of the populations studied, the types of treatments administered, and changes in the WHO guidelines for ART initiation over time. The WHO guidelines for minimal CD4+ T-cell counts at initiation changed from 200, to 350, to 500 and later to initiation at diagnosis. Thus, post-ART CD4+ T-cell growth rate in HIV-infected individual might be smaller now than it may have been on average in the past, as CD4+ T-cell counts at ART initiation are now higher.

## The effects of age

Baseline scaled CD4+ T-cell count was lower in adults compared to children (0.2 vs 0.8), that is, adults started treatment when their immune systems were more compromised compared to children. This could be due to the fact that children (<17 years old) are more likely vertically infected and, thus, they are more likely to be diagnosed early, owing to early testing and follow-up in the South African program for prevention of mother-to-child transmission. Adults, on the other hand, might be infected for an extended period and are consequently more likely to be more highly immune-compromised prior to diagnosis, compared to children.

We found that older age is associated with a lower value of baseline scaled CD4+ T-cell count in children and a higher value in adults. Vertical HIV- infection and an extended duration without treatment may lead to greater immune compromise in older children. In contrast, higher baseline scaled CD4+ T-cell counts in older adults suggest that they may have been more health conscious than younger adults (*Prohaska et al., 1985*), that is, younger adults tend to seek treatment later than older adults.

Such findings might also be explained by the negative correlation between post-ART CD4+ T-cell growth rates and baseline CD4+ T-cell counts: younger adults have a higher growth rate as they start at a lower CD4+ T-cell count value (*Cornell et al., 2012*). Thus, their CD4+ T-cell *rebound value*

may be higher than that of older adults over a similar period of time (*Means et al., 2016*; *International epidemiological Database to Evaluate AIDS (IeDEA) West Africa Collaboration et al., 2012*). Our results make no inferences regarding the period an individual takes to reach their normal CD4+ T-cell count value.

## Sex effects

We found that adult females have lower post-ART scaled carrying capacities than men, meaning they need a lower CD4+ T-cell count *rebound* to reach normality. This suggests that female adults tend to initiate treatment earlier than men (*Cornell and Myer, 2013*). This was also validated by the fact that baseline scaled CD4+ T-cell counts were higher in females than in males. A prior study has also found that women tend to have higher plateauing CD4+ T-cell counts than men (*Means et al., 2016*). In our study, women had lower rates of recovery than males which also supports our finding that higher baseline scaled CD4+ T-cell count is associated with a lower CD4+ T-cell growth rate. As they have higher CD4+ T-cell counts at ART initiation, and given the inverse correlation between the post-ART CD4+ T-cell growth rate and baseline scaled CD4+ T-cell count, women demonstrate lower growth rates.

Previous studies have found that adult males and females spend a similar time on therapy before reaching their rebound set-point CD4+ T-cell count (*EuroSIDA group et al., 2003*; *Patterson et al., 2007*). We do not consider this a contradiction of our results as an individual with a higher baseline CD4+ T-cell count will have a lower cell growth rate compared to an individual that started ART with a low CD4+ count. Thus, after an equivalent period following ART initiation similar increases in CD4 + T-cell counts may have occurred. We believe that the diverging opinions regarding the effects of sex on immune outcomes in adults post-ART initiation might be explained by variations in the definitions of 'immunological outcomes' by different authors and variations in analyses conducted.

This study has strengths and limitations. To ensure that patients had sufficient data to enable parameterization of the model, only subsets of the full data set were used in the analysis. This prevented over-fitting, but it might have introduced selection bias. However, with the exception of the percentage of people that suppressed viral load within 12 months of starting ART (*Table 1*), comparison of the summary statistics of the subset versus the full data set demonstrated that they were similar. Further, the results obtained from adjusting and not adjusting the model parameters with the variable, viral load suppression within 12 months, were also in agreement (see *Supplementary file 4*). We consequently believe that the subset used was representative of the full data set. Our model does not distinguish between naïve and memory sub-types of CD4+ T-cells. Prior studies have shown that these have different dynamics (*Di Mascio et al., 2006*; *Bajaria et al., 2002*). However, in routine HIV monitoring, for which we had data, the different subtypes are not measured. As ART treatment is for life, separating analyses for children from that for adults does not account for children growing into adulthood. Future studies might express the scaled carrying capacity as a function of age rather than as a variable parameter. Ignoring the thymus's contribution to CD4+ T-cell recovery, particularly in the earlier years of life, might have resulted in an overestimation of $r$ and $s$ parameters which are *aggregate* growth rates. Lastly, scaling the CD4+ T-cell counts of adults by a single average healthy CD4+ count value for adults might have introduced bias in the estimates of the scaled carrying capacity of healthy adults, due to variations of CD4+ T-cell counts across different age and demographic groups.

Our study does provide insight into the effect of ageing on immune system dynamics in adults and children on ART compared to healthy individuals. The ratio model provides a more accurate estimation of CD4+ counts reconstitution than the asymptotic model as well as the ability to compare different immune system outcomes, for both healthy and HIV-treated individuals. Using scaled CD4+ T-cell counts allows for the evaluation of CD4+ counts trajectories, which is not possible with unscaled CD4+ T-cell counts. We found large variations in CD4+ T-cell growth rates and scaled carrying capacities between individuals, highlighting the need to evaluate ART outcomes on an individual level. This calls for improved patient monitoring strategies. The strong inverse correlation between baseline scaled CD4+ T-cell count and the scaled carrying capacity emphasizes the importance of early ART initiation, regardless of age or state of disease progression. We found that post-ART CD4+ T-cell growth rate is not associated with a patient's age, but it is associated with higher baseline viral load. With the expansion of an aging population on ART, understanding long-term

effects of the treatment on the immune system is critical to ensure that the best care is delivered to HIV-infected patients.

## Acknowledgements

The authors gratefully acknowledge the support from Dr Joanna Lewis in the initial conceptualization of the study and the Centre for High Performance computing in Cape Town (http://www.chpc.ac.za) for providing part of the computational facilities.

## Additional information

### Funding

| Funder | Grant reference number | Author |
|--------|------------------------|--------|
| Schlumberger Stichting Fund, under the context of the Future Fellowship Program | | Eva Liliane Ujeneza |
| South African Department of Science and Technology and the National Research Foundation's Center of Excellence for Modelling and Analysis of Epidemiological Data | | Eva Liliane Ujeneza |
| National Institute Of Allergy And Infectious Diseases of the National Institutes of Health | U01AI069924 | Mary-Ann Davies |

The funders had no role in study design, data collection and interpretation, or the decision to submit the work for publication.

### Author contributions

Eva Liliane Ujeneza, Conceptualization, Data curation, Software, Formal analysis, Funding acquisition, Validation, Investigation, Visualization, Methodology; Wilfred Ndifon, Software, Formal analysis, Supervision, Validation, Investigation, Methodology; Shobna Sawry, Geoffrey Fatti, Julien Riou, Data curation; Mary-Ann Davies, Conceptualization, Data curation, Project administration; Martin Nieuwoudt, Conceptualization, Resources, Data curation, Supervision, Funding acquisition, Validation, Investigation, Methodology, Project administration

### Author ORCIDs

Eva Liliane Ujeneza ⓘ https://orcid.org/0000-0002-5760-4847
Geoffrey Fatti ⓘ http://orcid.org/0000-0002-6467-662X

### Ethics

Human subjects: This study was approved as part of the IeDEA Southern African collaboration's protocol, by the Human Research Ethics Committee of the University of Cape Town, with a reference number N1810119 RECIP UCT 084/2006. Informed consent was obtained from all participants by the clinics collecting the data according to IeDEA protocols.

### Decision letter and Author response

Decision letter https://doi.org/10.7554/eLife.42390.sa1
Author response https://doi.org/10.7554/eLife.42390.sa2

## Additional files

### Supplementary files

- Supplementary file 1. Range of the ratio model parameters.

- Supplementary file 2. Comparison of parameter estimates for adults, obtained using different fitting scenarios.
- Supplementary file 3. Evaluation of the variance–covariance matrix for the ratio and asymptotic models.
- Supplementary file 4. Additional estimated parameters.
- Transparent reporting form

## Data availability

Data used are from the International Epidemiologic Databases to Evaluate AIDS Southern Africa collaboration. They maintain a database of routinely collected data from various clinics, mostly located in South Africa. We recommend that interested readers contact Dr Morna Cornell, Project Manager IeDEA-SA in Cape Town (morna.cornell@uct.ac.za) to establish a data-sharing agreement. A research proposal highlighting how the data will be used is required. Source data for figures and figure supplements are provided, and the source code is available at https://github.com/EvaLiliane/RM_Code_eLife copy archived at https://archive.softwareheritage.org/swh:1:rev:624ff31c5fc969885f29b7291ee06886d24c64f7/.

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

## Appendix 1

### A1: Simulated references CD4+ T-cells values per age, for healthy children

Cross-sectional CD4+ T-cell counts data of 381 healthy children were collected as part of a study that sought to evaluate the use of international paediatric reference intervals by NHLS for South African populations (*Lawrie et al., 2015*). The study was conducted at a child wellness clinic, in Wesbank, a semi-informal settlement located in Cape Town. Healthiness was defined as not being on any prescribed medication, no chronic illness, a full clinical history, and attendance of the child to the clinic, with his/her biological mother. These children had a median age of 18.9 months IQR (6.5, 45.6) and a median CD4+ T-cell counts of 2007 cells/μL IQR (1433, 2749).

We simulated age-dependent normal CD4+ counts for healthy children, using a single exponential model as suggested in previous (unpublished) work. We model healthy CD4+ counts of a healthy child at age $a$ as follow:

$$f(a) = C_0 + C_1 \exp(-C_2 a),$$

where the constants $C_0$ and $C_1$ (unit: cells per μL) give an indication of the CD4+ count of a child at birth, and $C_2$ indicates the rate of decline of CD4+ counts (unit: cells per μL per month), thereafter. We used generalized nonlinear least squares method to estimate the model's parameters. We found that $C_0$, $C_1$, and $C_2$ were equal to 1070, 2.174, and 0.031 respectively. Using these values, we simulated CD4+ counts references values for each month of age, from birth until 17 years. As an indication, a sample of the simulated values are shown in the table below. The graph (see *Appendix 1—figure 1*) shows the raw data with the simulated references values as a red line.

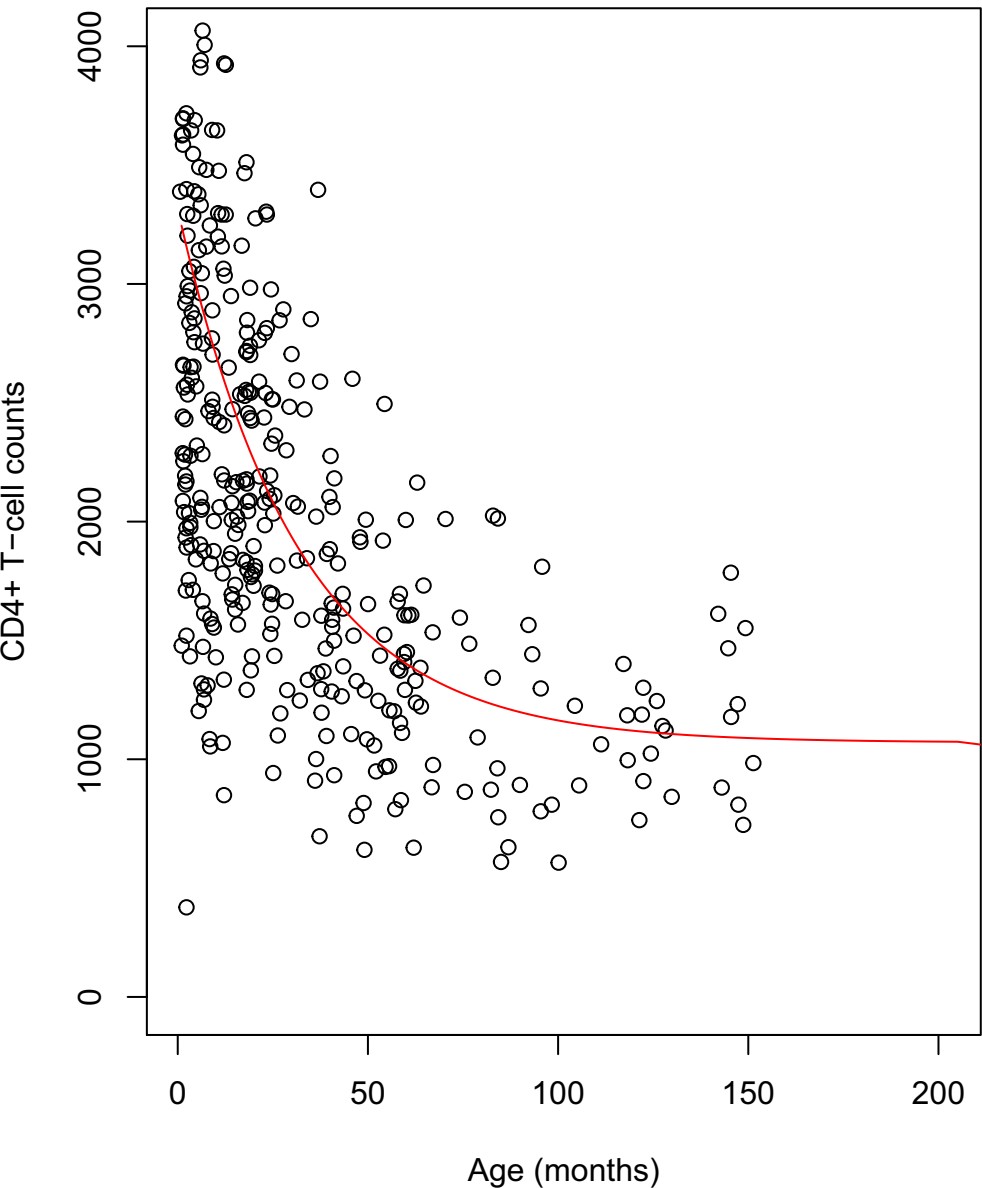

**Appendix 1—figure 1.** Plot of the simulated reference values for children. The dots represent the cross-sectional data for healthy children. The fitted red line shows the age-dependent reference values used in the scaling of CD4+ counts of HIV-infected children.

The online version of this article includes the following source data is available for figure 1:

**Appendix 1—figure 1—source data 1.** Simulated reference values for children.

**Appendix 1—table 1.** Sample of simulated healthy children's CD4+ count values.

| Age (months)/CD4+ T-cell counts/μL | | Age (years)/CD4+ T-cell counts/μL | | Age (years)/CD4+ T-cell counts/μL | | Age (years)/CD4+ T-cell counts/μL | | Age (years)/CD4+ T-cell counts/μL | |
|---|---|---|---|---|---|---|---|---|---|
| *0* | **3244** | *1* | **2554** | *5* | **1392** | *9* | **1108** | *13* | **1085** |
| *3* | 3046 | *2* | 2083 | 6 | 1290 | *10* | 1118 | 14 | 1080 |
| *6* | 2866 | *3* | 1761 | 7 | 1220 | *11* | 1102 | 15 | 1077 |
| *9* | 2703 | *4* | 1557 | *8* | 1172 | *12* | 1095 | 16 | 1075 |

## A2: Age and sex-dependent CD4+ T-cell counts reference values for healthy adults

In the study, we scaled all adults CD4+ counts by a single constant value, matching the studies below.

**Appendix 1—table 2.** Studies that evaluated normal CD4+ T-cells references ranges normal ranges in healthy African adult populations.

| Reference | Value (range) by gender | | Country | Age-related comments |
|---|---|---|---|---|
| | Male | Female | | |
| *Malaza et al., 2013*, IQR | 683 (542–849) | 833 (660–1038) | Durban, South Africa | Median age for women is 35 years vs 23 years for men. CD4+ count increase slightly with age till 64 years old |
| *Lawrie et al., 2009*, 2.5th and 97.5th percentile | (503–1807) | (561–2051) | Gauteng, South Africa | Average age of 41 years for all |
| *Ngowi et al., 2009*, mean ± sd | 665.6 ± 246.8 | 802 ± 250.2 | Tanzania | Mean age for women is 30.9 years vs 35.2 for men |
| *Lugada et al., 2004*, 5th and 95th percentile | 754 (362–1376) | 894 (454–1485) | Uganda | Decrease with age from birth to 18 years. All aged 24 were lumped together |
| *Institute of Human Virology/Plateau State Specialist Hospital AIDS Prevention in Nigeria Study Team et al., 2005*, mean ± sd | 838 ± 193 | 818 ± 213 | Nigeria | No age effect found |
| *Oladepo et al., 2009*, median | 746 (351–1455) | 892 (383–1654) | Nigeria | Decreasing with age |

We also fitted cubic splines to age-group published median CD4+ counts data (*Malaza et al., 2013*), for men and women separately. We then simulated age-dependent reference values as shown in the graph below (*Appendix 1—figure 2*). We found no difference between our initial results and those obtained from scaling data of HIV-infected adults by age and sex-dependent reference values.

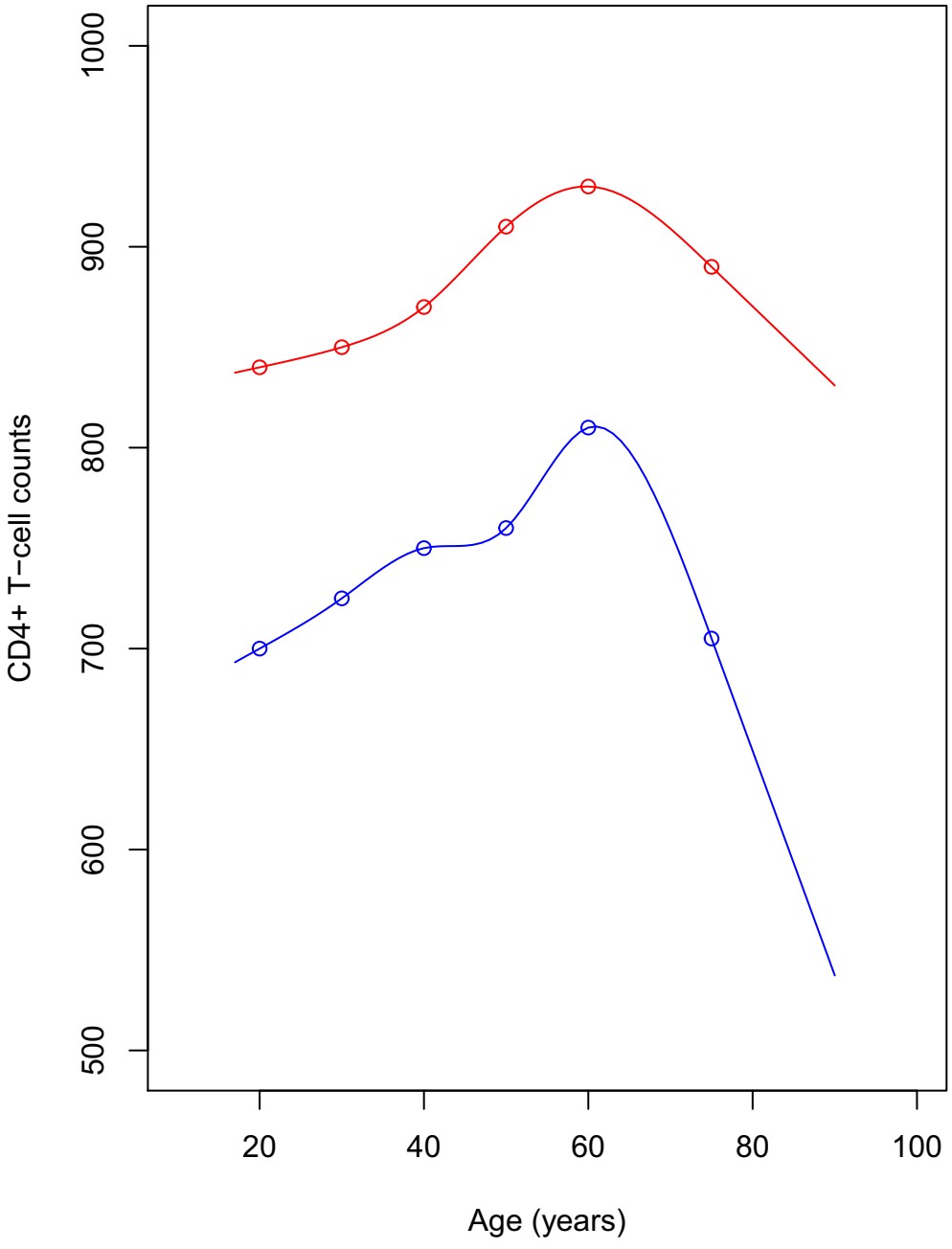

**Appendix 1—figure 2.** Plot of the simulated reference values for adults. The points represent the published median values. The red line shows the CD4+ count for women, blue line is for men. CD4+ reference values were simulated yearly, for ages ranging between 17 and 95.

The online version of this article includes the following source data is available for figure 2:

**Appendix 1—figure 2—source data 1.** Simulated reference values for adults.

