## [Decision Letter]

**Acceptance summary:**

HIV remains a major global health threat, with the virus infecting and destroying CD4^+^ (“helper”) T cells. This work combines clinical data and mathematical analysis to understand how T cell numbers are affected by infection and treatment at different ages.

**Decision letter after peer review:**

Thank you for submitting your article "A mechanistic model for long-term immunological outcomes in South African HIV-infected children and adults receiving ART" for consideration by *eLife*. Your article has been reviewed by two peer reviewers, and the evaluation has been overseen by a Reviewing Editor and Neil Ferguson as the Senior Editor. The following individual involved in review of your submission has agreed to reveal their identity: Nikos Pantazis (Reviewer #1).

The reviewers have discussed the reviews with one another and the Reviewing Editor has drafted this decision to help you prepare a revised submission.

Summary:

This manuscript describes a novel approach to characterizing CD4 cell recovery following ART initiation by modelling the ratio of CD4 counts of persons on ART to those of age/sex matched non-HIV infected individuals from the same population. The authors apply nonlinear mixed models which are based on plausible biological mechanisms. The model is demonstrated using treatment cohort data from children and adults from South Africa.

Both reviewers and I agree that the idea of modelling ratios is innovative and has great potential to improve the conceptualization of immune recovery following HIV treatment initiation. However, the impact of the work would be greatly enhanced through a clearer and more intuitive exposition and demonstration of both the proposed model and the alternative existing models which on which this work innovates. The reviewers have also raised important questions about the selection of individuals and data cleaning, and how these choices might influence the results. Finally, the data and assumptions for the “healthy individual” reference population in different scenarios need to be clarified. Most fundamentally, it is not clear how CD4 cell growth rates are estimates for healthy individuals when all of the data informing model inference are amongst ART patients.

We encourage resubmission of the manuscript, but anticipate that it will be substantially revised to provide an intuitive exposition of the proposed model.

Essential revisions:

– Incorporate an intuitive exposition of the proposed ratio model including figures illustrating the concepts and quantities introduced (carrying capacity, growth rate), how these are related to the model parameters, and how the model is affected by permuting each parameter.

– Consider focusing on only one of the two scenarios regarding the treatment of the baseline values as a covariate or not. Both options could be presented, explaining the pros and cons of each and how one was chosen, but then focus on one for the rest of the manuscript.

– Clearly explain how CD4 trajectories for non-HIV infected “healthy” individuals were modeled, and how these parameters were estimated given that there are no data about non-HIV infected individuals in the dataset used for model estimation. Subsection “Variable scaling” paragraph three describes CD4 is taken to be constant in healthy adults (independent of age and gender), but in the Results there is a growth rate for healthy adults >0. How is this growth rate estimated?

Given that CD4 levels at various ages come from external sources, authors should clarify better how they obtained the y-values required for the calculation of the ratios (z-values). Even reading S1, it is not clear if y-values for children of a given age were all the same (based on the predicted value from the exponential model; was it a double exponential as stated in the main text or single exponential as stated in S1?) or there was some variability allowed (and if yes, was it taken into account?). Moreover, I was puzzled even more in the adults case: how similar were the results when using modelled y-values (Figure 2—figure supplement 2) compared to using a constant value of 800 cells/μL? How different are the results compared to an analysis of just CD4 counts if scaling is just division by a constant?

– Relate the outputs of the models to outcomes of clinical relevance: such how long does it take to reach "normal" levels? What is the proportion of patients that is expected to reach these levels after a certain duration of treatment? How do baseline CD4, age, sex, viral load, virologic response etc. affect these quantities?

– Provide greater details on each of the exclusion criteria, how the selection of study participants might have affected estimates and results, and sensitivity analysis around these decisions. For example, more than 90% of the study participants are excluded when requiring at least 4 CD4 measurements and completeness in other key covariates. I suspect that the ">=4 CD4 measurements requirement" accounts for a large part of this >90% exclusion. What about the mechanisms behind this selection and their effects on the validity of the results? Do people have fewer CD4 measurements due to staggered entry into the study (more likely a "Missing Completely At Random" mechanism) or are they lost to follow-up or even dead before contributing at least 4 measurements (more likely "Missing At Random" or even "Missing Not At Random"). The issue should be thoroughly discussed and choices for the analysis should be well justified.

– Providing datasets and R code to explore the model and reproduce analyses may help readers to explore and understand the dynamics of the proposed model and data processing steps to prepare the modeled quantities.

[Editors' note: further revisions were suggested prior to acceptance, as described below.]

Thank you for resubmitting your work entitled "A Mechanistic Model for long-term Immunological outcomes in South African HIV-infected Children and Adults receiving ART" for further consideration by *eLife*. Your revised article has been evaluated by Neil Ferguson (Senior Editor) and a Reviewing Editor.

There are only a couple of remaining areas where we believe additional clarifications could be made. In particular, additions in the Materials and methods section and figures to give intuitive understanding of the model parameters would be helpful. Suggestions are outlined below:

Editorial comments:

– It is confusing that Equation 3 about CD4 trajectory for HIV-negative adults is expressed in terms of time since ART initiation since they do not initiate ART. It would be clearer to express this as a relationship between age and CD4 trajectory, and then expressing Equation 2 as a function of age at ART initiation and time since ART initiation.

– It would be helpful to give Equation 1 a descriptive interpretation of all of the parameters in Equations 2, 3, and 4 (k, q, y0, x0, z0, K, Q) and (2) examples of typical values or ranges for each parameter. It might be helpful to put this in a table that the reader can refer back to later.

– Subsection “Variable scaling”: My understanding of the methods is that the outcome variable that is modelled is the scaled CD4 count, rather than the CD4 count itself. Is that correct? If so, perhaps useful to clarify that in the first sentence of this section.

– Figure 2: This figure showing the logistic model would be more helpful for intuitively explaining the model with improved labelling and descriptions. What is “x-var” and “time-t”? What are the units on the axes? It is not clear how the vertical axis relates to CD4 or scaled CD4.

– Figure 3—figure supplement 2 and Figure 4—figure supplement 2: These figures are very helpful. A few aspects could be clearer:

* What are the solid and dashed lines?

* It would be helpful to add annotations for age at initiation and other relevant covariate values for each panel.

* Is it feasible to present this also with an axis for CD4 count? Or add a separate figure showing the data and trajectories for a couple respondents on both the scaled and natural CD4 scale?

– The e*Life* editorial guidance indicates that data and analysis code should be made available for tranparency and reproducibility. The authors have responded to the editorial request for this with contact information for a data request for IeDEA cohort data, but have not provided analysis code or data to reproduce the results. Especially given the stated objective of the manuscript to propose methodological advancement for modelling CD4 recovery, I think it would be very helpful to provide a limited dataset of the observations used in the analysis and code to reproduce the analysis, such that readers can implement and extend the proposed methods. This does not need to be the full IeDEA dataset, only the observations and covariates used in analysis in a suitable format for reproducing.

---

## [Author Response]

Essential revisions:– Incorporate an intuitive exposition of the proposed ratio model including figures illustrating the concepts and quantities introduced (carrying capacity, growth rate), how these are related to the model parameters, and how the model is affected by permuting each parameter.

We thank the reviewers for this suggestion. We added a figure illustrating the logistic growth model (Figure 2). We also modified the text to clarify the relationship between the parameters defined for the two logistic growth models and those defined for the ratio model, in addition to explanations that were already provided in the manuscript.

– Consider focusing on only one of the two scenarios regarding the treatment of the baseline values as a covariate or not. Both options could be presented, explaining the pros and cons of each and how one was chosen, but then focus on one for the rest of the manuscript.

We agree with the reviewers that focusing on a single scenario will increase the clarity of the manuscript. Therefore we revised the manuscript and focussed on scenario one, which consists of estimating all five parameters of the model. We also rewrote the Result section to reflect the modified presentation.

– Clearly explain how CD4 trajectories for non-HIV infected “healthy” individuals were modeled, and how these parameters were estimated given that there are no data about non-HIV infected individuals in the dataset used for model estimation. Subsection “Variable scaling” paragraph three describes CD4 is taken to be constant in healthy adults (independent of age and gender), but in the Results there is a growth rate for healthy adults >0. How is this growth rate estimated?

The model estimates parameter values (including the CD4^+^ growth rate in healthy individuals) that best fits the data. Despite the fact that for simplicity we set healthy individual CD4^+^ data to a constant value, the model predicts that a small growth rate for healthy individuals is required to explain the data. The fact that there is no identifiability issues in the estimates, and the consistency of the estimates obtained in both scenarios (using a constant healthy adults’ CD4 count of 800 cells/ul versus using age-dependent healthy adults’ CD4 counts) together suggest that the growth rate estimated in healthy individuals is correct. We have now clarified the text accordingly.

Given that CD4 levels at various ages come from external sources, authors should clarify better how they obtained the y-values required for the calculation of the ratios (z-values). Even reading S1, it is not clear if y-values for children of a given age were all the same (based on the predicted value from the exponential model; was it a double exponential as stated in the main text or single exponential as stated in S1?) or there was some variability allowed (and if yes, was it taken into account?).

We thank the reviewers for this comment. The model used in simulating reference CD4^+^ values for healthy children is a single exponential model. We have now clarified this in the text. Regarding the y-values used for children of different ages, we used a single simulated average value per age-month group. Given that these were external data and that CD4^+^ values have a large variability between individuals, we decided that it is best to use average values.

Moreover, I was puzzled even more in the adults case: how similar were the results when using modelled y-values (Figure 2—figure supplement 2) compared to using a constant value of 800 cells/μL?

We found that 95% confidence intervals for population-level parameters estimates obtained using a constant healthy adults’ CD4 count of 800 cells/ul overlapped with those obtained using age-dependent reference values, indicating there is no significant difference, in all cases except for the parameter z_0_, whose estimate was higher in the latter scenario (please see Supplementary file 1). Nevertheless, the percent relative difference between the estimates for z_0_ was only 18.6% (Supplementary file 1). The percent relative difference between estimates for all the other parameters was less than 10% (Supplementary file 1).

Similarly, the 95% confidence intervals for individual-level parameters obtained using a constant healthy adults’ CD4 count of 800 cells/ul overlapped with those obtained using age-dependent reference values, indicating there is no significant difference, in all cases (Supplementary file 1). Moreover, the percent relative difference between the parameters estimates was less than 10% in all cases (Supplementary file 1).

Together, these observations reinforce our confidence that the parameter estimates obtained for adults are not affected by the scaling of CD4^+^count. We have now cited these observations in the text.

How different are the results compared to an analysis of just CD4 counts if scaling is just division by a constant?

Only two parameters in the ratio model (namely z_0_ and Q) are directly scaled by baseline data from healthy individuals. Among these two parameters, we find that the lack of scaling markedly affects the population-level and individual-level estimates of only z_0_ (please see Supplementary file 1). Strikingly, compared to the case when CD4 counts of HIV patients are scaled by a constant value of 800 cells/ul, in the absence of such baseline scaling the mean estimates estimate for z0 increase by a factor about 800 (Supplementary file 1). In contrast, estimates for all other parameters are much less affected by the lack of baseline scaling (Supplementary file 1). We have now cited these results in the text.

– Relate the outputs of the models to outcomes of clinical relevance: such how long does it take to reach "normal" levels? What is the proportion of patients that is expected to reach these levels after a certain duration of treatment? How do baseline CD4, age, sex, viral load, virologic response etc. affect these quantities?

As suggested by the reviewers, we added text to the manuscript to clarify the clinical relevance. We explain the time it takes to reach the threshold of “500 CD4^+^ T-cell counts” for both adults and children. We also added/gave an indication of the proportion of people in our dataset, which actually reached the 500 CD4^+^ T-cell counts threshold, within a 12 and 24 months period spent on treatment.

– Provide greater details on each of the exclusion criteria, how the selection of study participants might have affected estimates and results, and sensitivity analysis around these decisions. For example, more than 90% of the study participants are excluded when requiring at least 4 CD4 measurements and completeness in other key covariates. I suspect that the ">=4 CD4 measurements requirement" accounts for a large part of this >90% exclusion. What about the mechanisms behind this selection and their effects on the validity of the results? Do people have fewer CD4 measurements due to staggered entry into the study (more likely a "Missing Completely At Random" mechanism) or are they lost to follow-up or even dead before contributing at least 4 measurements (more likely "Missing At Random" or even "Missing Not At Random"). The issue should be thoroughly discussed and choices for the analysis should be well justified.

As pointed out by the reviewers, the majority of excluded individuals did not have enough CD4^+^ T-cell count observations, to allow for the estimation of a unique set of parameters. Given that we did use data that was routinely collected as part of broad HIV/AIDS regional program, we were expecting this to happen. Various factors account for missing data in our study cohorts. For instance, a patient transferring from another facility to another while on ART will be assigned a new unique identifier, and his/her data will be considered as coming from two different individuals in our study. If both set of data have less than the minimum required for our analysis, this patient will be excluded. Some patients also died whereas others were lost-to-follow up (Table 1).

One of the motivation of this study was to develop a technique that can be applied to very noisy data, and still be useful. Thus, we started with an enormous dataset because we know a lot would get excluded. We evaluated the differences in the fixed parameters estimated for the Asymptotic model, for patients with a 3, 4 and 5 minimum observations. We found no systematic differences in the estimated parameters, as we restricted the analysis to patients with more observations (see Author response table 1).

**Author response table 1. resptable1:** 

	Asymptote	Log of the rate of increase	Intercept	BIC
Min 3 observations*# 17241 adults*	0.632*CV 0.70%*	0.071*CV 1.88%*	0.142*CV 0.77%*	-148917
Min 4 observations*14,542*	0.58CV 0.63%	0.11*CV 2.0%*	0.14*CV 0.81%*	-136244
Min 5 observations*12,238*	0.66*CV 0.77%*	0.06*CV 2.11%*	0.14*CV 0.91%*	-126716

– Providing datasets and R code to explore the model and reproduce analyses may help readers to explore and understand the dynamics of the proposed model and data processing steps to prepare the modeled quantities.

Data are from the International epidemiologic Databases to Evaluate AIDS Southern Africa collaboration. We added a statement on the process to acquire it/them in the paper.

[Editors' note: further revisions were suggested prior to acceptance, as described below.]

Editorial comments:– It is confusing that Equation 3 about CD4 trajectory for HIV-negative adults is expressed in terms of time since ART initiation since they do not initiate ART. It would be clearer to express this as a relationship between age and CD4 trajectory, and then expressing Equation 2 as a function of age at ART initiation and time since ART initiation.

We thank the reviewer for these comments. Equation (3) does not describe CD4 evolution from birth for healthy patient, but their evolution starting at the age they are being matched to an HIV-infected patient. Thus, expressing the model as a function of age, would be inconsistent with our assumptions. We agree with the reviewer that this might be a bit confusing to the reader. Thus, we have now added the below sentence to clarify the text.

“Thus, the equation describes the immune system dynamics of the healthy individual, starting from when they have a similar age as their corresponding HIV-infected counterpart.”

– It would be helpful to give Equation 1 a descriptive interpretation of all of the parameters in Equations 2, 3, and 4 (k, q, y0, x0, z0, K, Q) and (2) examples of typical values or ranges for each parameter. It might be helpful to put this in a table that the reader can refer back to later.

A descriptive interpretation of parameters *k, r* and *x0* are available within the section “Ratio” model construction, and a visual representation is available in Figure 2. The two other parameters q and y0, for healthy individual, are also described.

For additional clarity, as requested by the reviewer, we have now added a table with the expected range for each of the mentioned parameters (see supplementary material 1).

– Subsection “Variable scaling”: My understanding of the methods is that the outcome variable that is modelled is the scaled CD4 count, rather than the CD4 count itself. Is that correct? If so, perhaps useful to clarify that in the first sentence of this section.

As the reviewer points out, *scaled CD4* is indeed the outcome variable, as mentioned in the first sentence of the section “Variable Scaling”. For added emphasis, we have added “to obtain the outcome variable” to the end of the second sentence of the same section.

– Figure 2: This figure showing the logistic model would be more helpful for intuitively explaining the model with improved labelling and descriptions. What is “x-var” and “time-t”? What are the units on the axes? It is not clear how the vertical axis relates to CD4 or scaled CD4.

The logistic model is used to describe the dynamics of CD4 counts. Therefore, following the reviewer’s advice, we have now changed the label for the y-axis and modified the description in the legend to improve clarity (see Figure 2).

– Figure 3—figure supplement 2 and Figure 4—figure supplement 2: These figures are very helpful. A few aspects could be clearer:* What are the solid and dashed lines?* It would be helpful to add annotations for age at initiation and other relevant covariate values for each panel.

We thank the reviewer for these comments. We have added individual information to each panel of both figures and modified the legend to improve clarity (see Figure 3—figure supplement 2 and Figure 4—figure supplement 2).

* Is it feasible to present this also with an axis for CD4 count? Or add a separate figure showing the data and trajectories for a couple respondents on both the scaled and natural CD4 scale?

For adults, given that CD4 count measurements were scaled with a constant, the graph of their trajectories would be like that of the scaled CD4 count, with the only difference being the scale for the y-axis. Thus, we chose not to show these graphs.

– The eLife editorial guidance indicates that data and analysis code should be made available for tranparency and reproducibility. The authors have responded to the editorial request for this with contact information for a data request for IeDEA cohort data, but have not provided analysis code or data to reproduce the results. Especially given the stated objective of the manuscript to propose methodological advancement for modelling CD4 recovery, I think it would be very helpful to provide a limited dataset of the observations used in the analysis and code to reproduce the analysis, such that readers can implement and extend the proposed methods. This does not need to be the full IeDEA dataset, only the observations and covariates used in analysis in a suitable format for reproducing.

Given our data sharing agreement with IeDEA consortium, we unfortunately cannot share any part of the dataset we received. However, provided randomly generated data, that are not derived to the actual dataset used in the analysis but contains similar information. This also means that whoever uses it might not necessary obtained the exact same output parameters as that we have. Below is the edited paragraph on Data Sharing:

“Data for HIV-infected patient are from the International epidemiologic Databases to Evaluate AIDS Southern Africa collaboration. We recommend that interested readers contact Dr. Morna Cornell, Project Manager IeDEA-SA in Cape Town (morna.cornell@uct.ac.za) to establish a data sharing agreement. Simulated data for healthy adults and children are available through the *eLife* data sharing platform.”

We have also provided a link to the folder containing the analysis code (the link to be added at submission). Initial parameters used in the analysis will be part of the code.